# ViewMask-1-to-3: Multi-View Consistent Image Generation via Multimodal Discrete Diffusion Models

**Ruishu Zhu** [1 2 * †] **Zhihao Huang** [1 2 * †] **Jiacheng Sun** [3] **Ping Luo** [4] **Hongyuan Zhang** [2 4] **Xuelong Li** [2]

## Abstract

Motivated by discrete diffusion's success in language-vision modeling, we explore its potential for multi-view generation, a task dominated by continuous approaches. We introduce **ViewMask-1-to-3**, formulating multi-view generation as a discrete sequence modeling problem where each viewpoint is represented as visual tokens from MAGVIT-v2. Through **discrete diffusion** via masked token prediction, our approach **enables progressive multi-view generation via iterative token unmasking**, unifying language and vision in a shared token space. Importantly, simple random masking combined with self-attention naturally encourages cross-view consistency without specialized architectures or 3D geometric priors. Our method outperforms the baseline on the GSO and 3D-FUTURE benchmarks, ranking first on average across standard image metrics, and achieving a 10.6% higher IoU than continuous diffusion models on 3D-FUTURE. Furthermore, the proposed framework can be naturally extended to support text-to-image generation and multimodal understanding, highlighting its potential toward a more unified paradigm for multimodal understanding and generation.

## 1. Introduction

Multi-view image generation has emerged as a fundamental challenge (He et al., 2024b; Liu et al., 2024b; He et al., 2024a), with applications spanning virtual reality (Zhou et al., 2025) and 3D reconstruction (Alzayer et al., 2024). The task aims to generate multiple consistent viewpoints of an object or scene from limited input—typically a single image and accompanying text description. Computational approaches face significant challenges in maintaining geometric consistency, preserving fine-grained details, and ensuring semantic coherence across viewpoints.

Recent advances in this domain have achieved impressive results through diverse technical approaches. **3D representation methods** (Xiang et al., 2025; Tang et al., 2024; Liu et al., 2024c) leverage explicit 3D representations, such as neural radiance fields, 3D Gaussian primitives, or structured latent spaces, to ensure geometric consistency through native 3D modeling and rendering. **Camera-conditioned diffusion models** (Liu et al., 2023b; Shi et al., 2023; Kwak et al., 2024; Huang et al., 2024) have demonstrated remarkable flexibility by directly generating multi-view images in 2D space, either controlling the views with precise camera parameters or directly producing fixed views. **Image Editing model** (Huang et al., 2025; Long et al., 2024; Liu et al., 2023a) explore integrating multi-view generation with image editing manner, enabling seamless transitions between different vision tasks. These methods, predominantly built on continuous diffusion in latent space, have collectively established strong baselines.

Meanwhile, the rapid advancement of multimodal understanding (Radford et al., 2021; Hu et al., 2026; Zhu et al., 2026; Li, 2022; Zhang et al., 2025a; An et al., 2026) and generation (Esser et al., 2024; Wang et al., 2024; Gu et al., 2026; Huang et al., 2026) has driven a growing trend toward unified modeling of both perception and synthesis within a single framework (Wu et al., 2025a; Xie et al., 2025b). Within this trend, **discrete diffusion models based on masked token prediction** (Nie et al., 2026; You et al., 2025; Yang et al., 2026; Shi et al., 2025) have shown remarkable success in multimodal understanding and generation. This architecture offers several advantages: (1) faster inference than autoregressive models (You et al., 2025; Xin et al., 2025a), (2) native fusion of text and visual tokens, and (3) natural alignment with large language models. This raises an unexplored question: *Can this discrete paradigm serve as a viable alternative for the geometrically demanding task*

---

*Equal contribution †Work done during an internship at TeleAI.
[1]School of Artificial Intelligence, OPtics and ElectroNics (iOPEN), Northwestern Polytechnical University [2]Institute of Artificial Intelligence, China Telecom (TeleAI) [3]Huawei Foundation Model Department [4]The University of Hong Kong. Correspondence to: Hongyuan Zhang <hyzhang98@gmail.com>, Xuelong Li <xuelong_li@ieee.org>.

*Proceedings of the 43rd International Conference on Machine Learning*, Seoul, South Korea. PMLR 306, 2026. Copyright 2026 by the author(s).

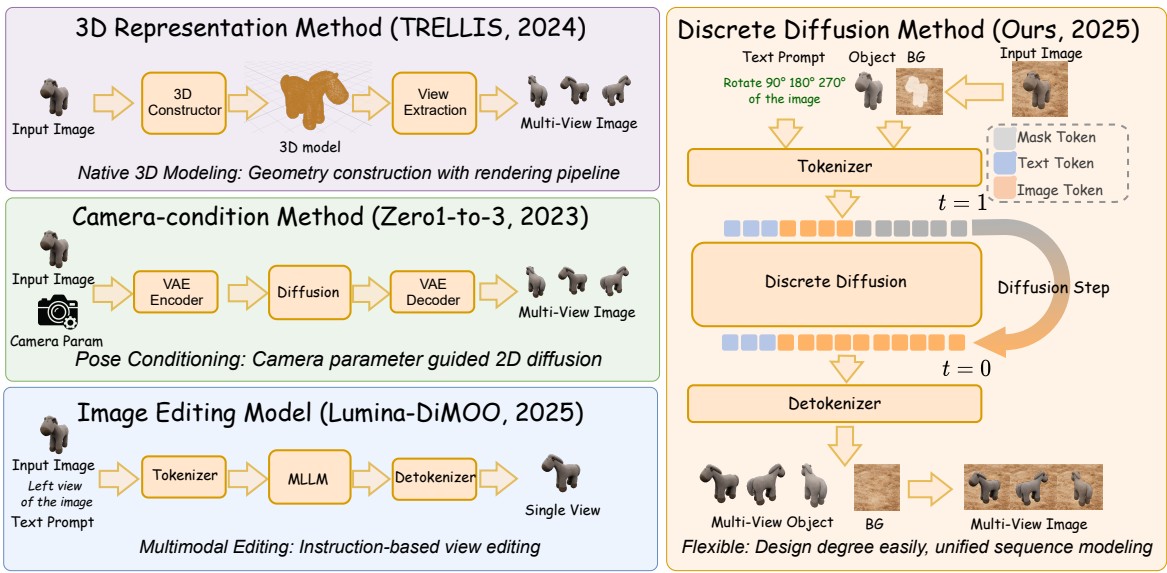

*Figure 1.* Comparison of multi-view image generation approaches. (Left-Top) 3D representation methods with 3D reconstruction and rendering. (Left-Bottom) Camera-conditioned diffusion methods with precise camera parameters. (Left-middle) Image editing models generate views with sequential manner. (Right) Our discrete diffusion approach enables parallel multi-view generation through unified sequence modeling with flexible text-based control.

*of multi-view generation?*

In this work, we present ViewMask-1-to-3, an exploration of discrete diffusion models for multi-view consistent image generation. Our key insight is to formulate multi-view generation as a *discrete sequence modeling problem*, where each viewpoint is represented as a sequence of visual tokens obtained through learned tokenization (Yu et al., 2024). By extending masked token prediction, a technique proven effective in language modeling (Nie et al., 2026), to the visual domain with multiple viewpoints, we enable progressive generation through iterative token unmasking. Importantly, we find that a simple random masking strategy, when combined with bidirectional self-attention, naturally encourages cross-view consistency without requiring specialized architectural components or explicit 3D geometric priors. Our contributions are as follows:

- We present a systematic exploration of discrete diffusion models for multi-view image generation, demonstrating that masked token prediction can achieve competitive performance in this domain.

- The proposed simple masking strategy, when combined with bidirectional attention mechanisms, naturally encourages cross-view consistency without requiring 3D geometric priors or specialized architectures.

- We evaluate ViewMask-1-to-3 on standard multi-view generation benchmarks, where it tops the average rank-

ings on image-level metrics (PSNR, SSIM, LPIPS, CD, IOU) for GSO and 3D-FUTURE. In particular, compared with continuous diffusion models, it achieves a 10.6% higher IoU on 3D-FUTURE.

## 2. Related Work

### 2.1. Multi-View Generation

**3D Representation-Based Methods**  Early approaches ensure geometric consistency through explicit 3D modeling. 3D-aware GANs such as $\pi$-GAN (Chan et al., 2021) and EG3D (Chan et al., 2022) leverage neural radiance fields or tri-plane representations for view-consistent synthesis. More recently, DreamGaussian (Tang et al., 2024) employs 3D Gaussian Splatting for efficient content creation; TRELLIS (Xiang et al., 2025) adopts flow matching to obtain structured 3D latents and then extracts multi-view images; MV-Dream (Shi et al., 2024) and SyncDreamer (Liu et al., 2024c) integrate NeRF (Mildenhall et al., 2021) into diffusion frameworks with denoising across viewpoints.

**Camera-Conditioned Diffusion Methods**  An alternative approach generates multi-view images directly in 2D space through camera-conditioned diffusion. Zero-1-to-3 (Liu et al., 2023b) pioneered fine-tuning pretrained diffusion models with camera pose conditioning for zero-shot novel view synthesis, generating each view independently. Subsequent works address cross-view consistency through various

strategies: Zero123++ (Shi et al., 2023) models the joint distribution of multiple views by concatenating them into a unified representation; SyncDreamer (Liu et al., 2024c) and One-2-3-45 (Liu et al., 2023a) incorporate synchronized denoising and post-processing refinement. Recent extensions include ViVid-1-to-3 (Kwak et al., 2024), which leverages video diffusion for temporal coherence; MV-AR (Hu et al., 2025), which introduces autoregressive viewpoint generation; and EpiDiff (Huang et al., 2024), which enforces epipolar constraints through specialized attention mechanisms. However, these methods typically focus on image-to-multi-view synthesis, with text-to-multi-view requiring additional text-to-image models to generate reference images first.

**Image Editing Methods**  Recent methods attempt to generate multi-view images via image editing strategies. These approaches leverage various techniques: MV-Adapter (Huang et al., 2025) transforms text-to-image models into multi-view generators without altering their structure; ImageBrush (Yang et al., 2023) employs visual instructions rather than text for precise manipulation; SceneScape (Fridman et al., 2023) ensures geometric consistency through depth-aware generation; and Lumina-DiMOO (Xin et al., 2025a) introduces discrete diffusion in image editing with impressive results. While these methods rely on specialized editing strategies, ViewMask-1-to-3 naturally achieves cross-view consistency through token-level modeling using discrete diffusion, offering a more unified and elegant solution to multi-view synthesis.

### 2.2. Discrete Diffusion Models

D3PM (Austin et al., 2021) established the foundational framework for discrete diffusion, extending traditional diffusion to discrete state spaces through structured denoising processes. DiffusionBERT (He et al., 2023) demonstrated that pre-trained BERT models could be adapted for text generation through discrete diffusion training. A significant breakthrough came with LLaDA (Nie et al., 2026), which showed that masked diffusion models can match autoregressive models like LLaMA3-8B in language generation while offering parallel decoding and bidirectional context modeling. LLaDA-V (You et al., 2025) extended this framework to multimodal understanding, demonstrating the potential of discrete diffusion in vision-language tasks. These advances have established discrete diffusion as a viable alternative to autoregressive approaches, for tasks requiring controllable and parallel generation.

### 2.3. Visual Tokenization

The effectiveness of discrete diffusion for visual tasks critically depends on visual tokenization quality. VQ-VAE (van den Oord et al., 2017) introduced discrete visual representations through vector quantization, laying the

groundwork for discrete visual modeling. MaskGIT (Chang et al., 2022) applied masked token prediction to image generation using vector-quantized representations. MAGVIT-v2 (Yu et al., 2024) enables high-quality discrete tokenization for both images and videos through lookup-free quantization and architectural improvements. The success of these tokenizers make discrete approaches increasingly competitive with continuous methods in visual generation tasks.

## 3. Method

### 3.1. Overview

Given a single input image $I_0 \in \mathbb{R}^{H \times W \times 3}$ and an optional text description $T$, our goal is to generate $N = 3$ consistent target views $\{I_1, I_2, I_3\}$ representing the same object from different viewpoints. Beyond image-conditioned view synthesis, our model also achieves text-only generation: provided solely with a description $T$, it can synthesize $N = 4$ new views $\{I_1, I_2, I_3, I_4\}$ that correspond to distinct viewpoints, even in the absence of an input image. This unified formulation enables both image-to-multi-view and text-to-multi-view generation within a single framework. Unlike Zero-1-to-3 that operates in continuous latent spaces, we formulate multi-view generation as a discrete sequence modeling problem. By representing each view as a sequence of discrete visual tokens, we can **leverage the parallel processing and bidirectional context modeling capabilities of masked diffusion models**.

### 3.2. Model Design

Our approach consists of three main components: (1) Visual Tokenization, where we convert images to discrete token sequences using MAGVIT-v2; (2) Cross-View Sequence Construction, where we build unified sequences encoding multiple viewpoints with special tokens; and (3) Masked Diffusion Training, where we learn to predict masked visual tokens through iterative refinement.

**Visual Tokenization**  We adopt MAGVIT-v2 (Yu et al., 2024) as our visual tokenizer due to its superior performance in discrete visual representation. (More discussions can be found in Appendix C) For an input image $I_i$, the tokenizer produces a sequence of discrete visual tokens,

$$V^{(i)} = \text{Tokenize}(I_i) = \{v_1^{(i)}, v_2^{(i)}, \ldots, v_L^{(i)}\}, \quad (1)$$

where $V^{(i)} \in \mathcal{V}^L$ represents the token sequence, $\mathcal{V}$ is the visual vocabulary with $|\mathcal{V}| = 2^{18}$ tokens, and $L$ is the sequence length. Each token $v_j^{(i)}$ corresponds to a spatial region of the image and encodes local visual features. The discrete representation enables us to apply language modeling techniques directly to visual content while maintaining reconstruction quality through the learned tokenizer.

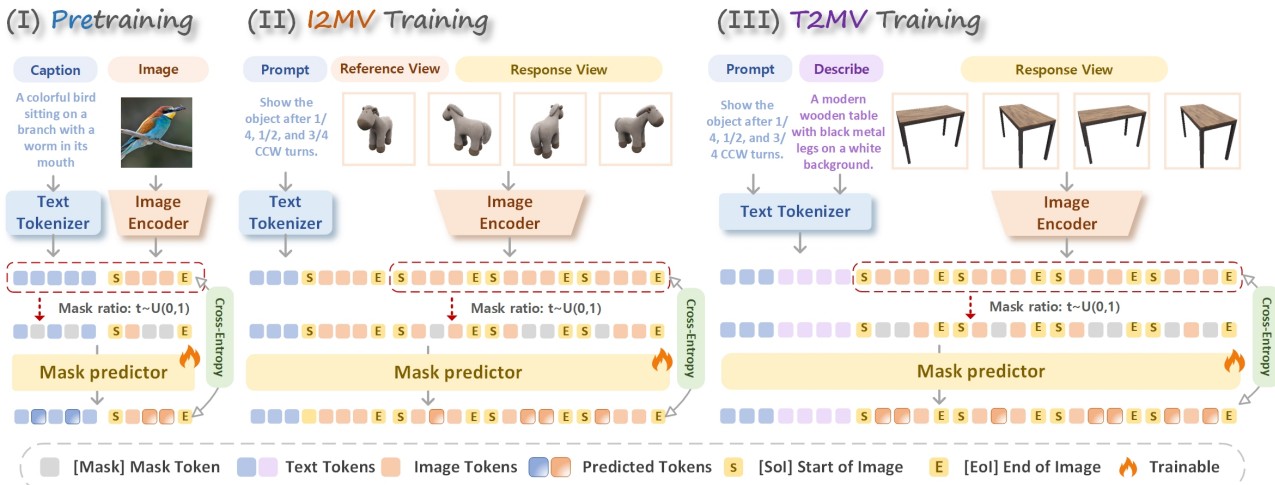

*Figure 2.* **Overview of our three-stage training framework. (I) Stage 1**: Pretraining aligns text and image modalities using masked image-caption sequences. **(II) Stage 2**: Image-to-Multi-View (I2MV) masks the target view and learns to reconstruct it conditioned on a reference view, thereby acquiring multi-view generation capability. **(III) Stage 3**: Text-to-Multi-View (T2MV) masks all views and learns to generate multiple views of an object conditioned on a text description.

The tokenized sequences are then embedded into the language model's embedding space:

$$\mathbf{e}^{(i)} = \text{Embed}(V^{(i)}) \in \mathbb{R}^{L \times d}, \quad (2)$$

where $d$ is the embedding dimension of the transformer.

**Cross-View Sequence Construction**  The unified sequence construction is a core step of ViewMask-1-to-3, which enables coordinated generation across multiple viewpoints under both image-to-multi-view (I2MV) and text-to-multi-view (T2MV) settings. By using the same masked-sequence format, we minimize the structural gap between I2MV and T2MV inputs, enabling the model to treat both tasks similarly (see Section 3.3 for details).

**Cross-View Masking Strategy**  During training, we employ a simple yet effective random masking strategy. For each training sample, target tokens are first selected based on the current training stage (see Section 3.3), and a masking ratio $r \sim \text{Uniform}(0, 1)$ is sampled to randomly replace a subset of them with a special **[MASK]** token.

### 3.3. Multi-Stage Training Strategy

As shown in Fig. 2, our training framework comprises three stages.

**Stage1: Pretraining for Multimodal Alignment**  To enable the LLADA model to learn representations of image tokens, we leverage image-caption pairs to align the two modalities. The input sequence is formatted as:

[T2I] [SOT] *caption tokens* [EOT] [SOI] *image tokens* [EOI],

where [T2I] indicates the image-caption alignment, [SOT] and [EOT] denote the start and end of the text, and [SOI] and [EOI] denote the start and end of the image. Both *caption tokens* and *image tokens* are randomly masked with a certain probability during training. This design encourages the model to fully capture the co-occurring semantic information in image–caption pairs, thereby aligning the representations of the two modalities.

We construct a multimodal pretraining corpus of 1.2 million image–caption pairs using the images from ImageNet (Deng et al., 2009) images combined with captions provided by imagenet-1k-vl-enriched (Layer, 2024), to facilitate alignment between modalities.

**Stage2: Image to Multi-View**  Building on the multimodal alignment from Stage 1, we fine-tune the model on image-to-multi-view generation. The input sequence is formatted as:

[I2MV] [SOT] *P.* [EOT] [SOI] *R.* [EOI] [SOI] $G_1\ G_2\ G_3$ [EOI],

where [I2MV] indicates the image-to-multi-view task, *P.* represents the prompt tokens, *R.* represents the reference image tokens, and $G_i$ represents the generated image tokens. For brevity, the repeated [EOI] [SOI] between consecutive $G_i$ are omitted. $G_i$ are randomly masked with a certain probability during training.

At this stage, the model is trained on a curated subset of Objaverse (Deitke et al., 2023) and Habitat Synthetic Scene Dataset (HSSD) (Khanna et al., 2024). Following the data split provided by Objaverse++ (Lin et al., 2025), we removed samples labeled as Low Quality (No seman-

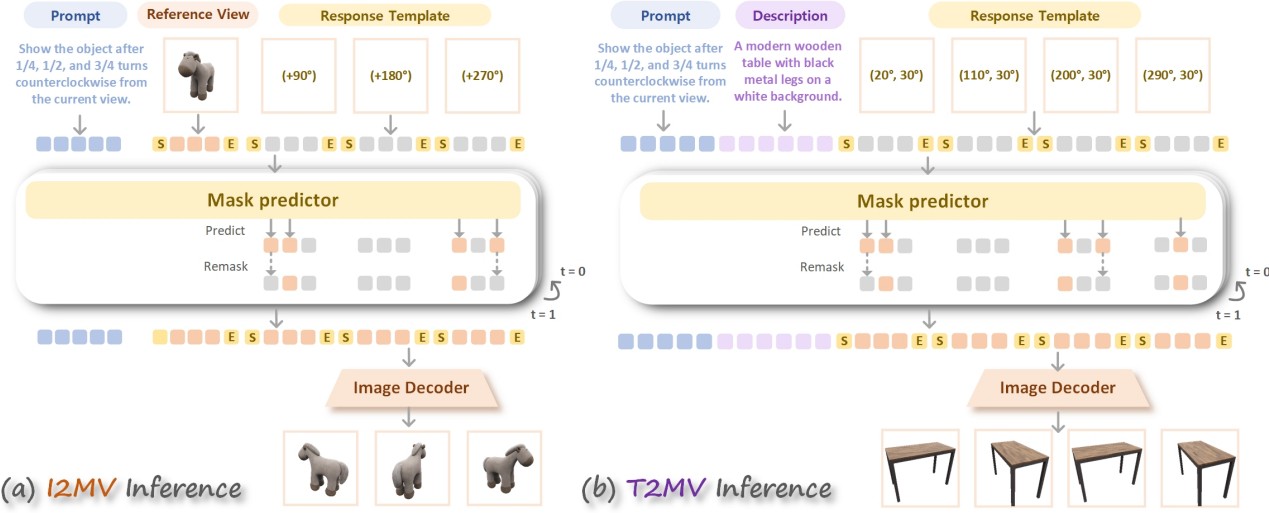

*Figure 3.* **Overview of inference framework. (a) I2MV**: The tokenized prompt and reference view provide a template for generating three consistent output images from different viewpoints. **(b) T2MV**: Tokenized prompt and description, providing a template for generating four output images. At each timestep t, the tokens are further predicted, and under the guidance of the mask schedule, remasking is performed based on confidence.

tic meaning. Objects that annotators cannot identify or are corrupted), to ensure more stable training.

**Stage3: Text to Multi-View** To extend the I2MV paradigm to T2MV, we incorporate an additional textual description specifying the target views immediately after the prompt. Accordingly, the input sequence is formatted as:

$$[\text{T2MV}]\ [\text{SOT}]\ P.\ D.\ [\text{EOT}]\ [\text{SOI}]\ G_1\ G_2\ G_3\ G_4\ [\text{EOI}],$$

where [T2MV] indicates the text-to-multi-view task, $D.$ represents the description tokens. For brevity, the repeated [EOI] [SOI] between consecutive $G_i$ are omitted. $G_i$ are randomly masked with a certain probability during training.

We enhance Objaverse with Cap3D (Luo et al., 2023; 2024), which offers detailed textual descriptions for 3D assets.

**Loss** ViewMask-1-to-3 is trained using a cross-entropy loss that predicts original tokens at masked positions.

$$\mathcal{L}_{\text{CE}} = -\sum_{i=1}^{3} \sum_{j \in \mathcal{M}_i} \log P(v_j^{(i)} | s_{\backslash \mathcal{M}}), \qquad (3)$$

where $\mathcal{M}_i$ denotes the set of masked positions in view $i$, and $s_{\backslash \mathcal{M}}$ represents the sequence with masked tokens. This objective encourages the model to reconstruct original visual tokens while leveraging unmasked context from the input image, text description, and partially observed target views. The cross-attention mechanism naturally enables information flow between different modalities and viewpoints.

### 3.4. Inference Process

**Construction of Input Sequences** During inference, we perform iterative denoising starting from fully masked target views, as illustrated in Fig. 3. For **I2MV**, the model is provided with a tokenized prompt and the reference view. The template is designed to generate three output images. For **T2MV**, the input consists of a tokenized prompt and a textual description of the desired view. This template supports the generation of four output images.

**Predict and Remask** We employ an iterative denoising-based sampling strategy. The model progressively predicts token distributions over $T$ iterations. At each step, the model performs a forward pass to obtain logits for all masked positions and estimates the confidence of each token prediction. A scheduling function (cosine, linear, or quadratic) determines the number of tokens to be re-masked in the next step, where low-confidence tokens are re-masked while high-confidence ones are retained. This *predict–re-mask* loop continues until all masked tokens are generated, progressively refining the sequence toward the final output. We further discuss the effects of different scheduling strategies in the ablation study (see Section 4.3).

## 4. Experiments

**Datasets** Our model was trained on a subset of Objaverse (Deitke et al., 2023) and HSSD (Khanna et al., 2024). Following the data split provided by Objaverse++ (Lin et al., 2025), we removed samples labeled as Low Quality, to ensure more stable training. To enrich the textual supervision,

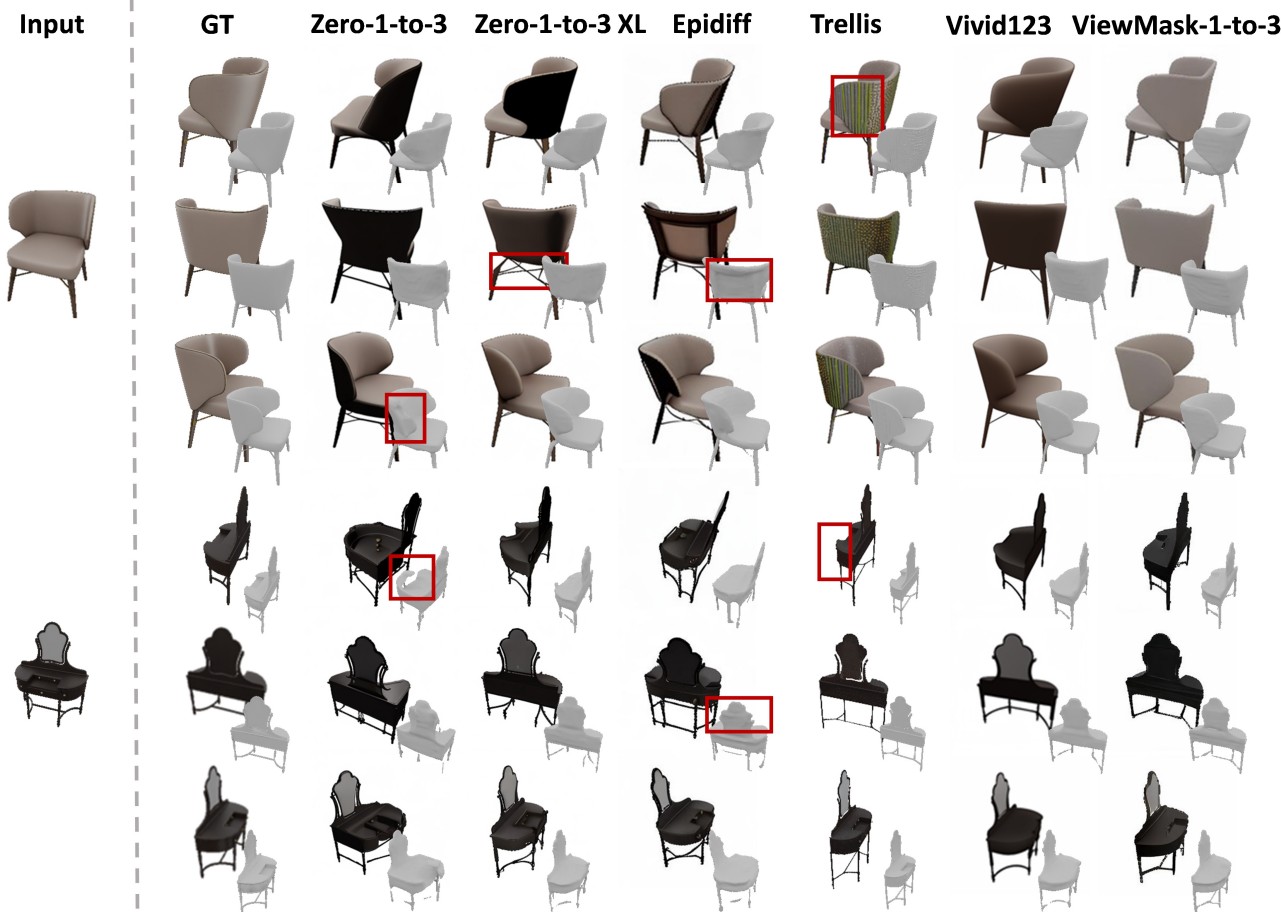

*Figure 4.* **Qualitative comparison on the 3D-Future dataset.** Visual results on the 3D-Future dataset, where the bottom-right shows the reconstructed mesh, demonstrating that our ViewMask-1-to-3 produces more consistent and realistic novel views compared with recent methods.

we adopted the Cap3D (Luo et al., 2023; 2024) annotations, which provide descriptive captions for 3D objects. In total, we used 180K 3D objects, each rendered as an 8-frame RGBA orbit sequence at a resolution of 256×256, with the elevation fixed at 30°. During training, one frame was randomly selected as the reference image for each object, resulting in a multi-view generation dataset. (More discussions can be found in Appendix A)

**Metrics** We evaluate the multi-view generation results from two perspectives. First, for the directly generated novel views, we adopt common image-level metrics, including Peak Signal-to-Noise Ratio (PSNR), Structural Similarity Index Measure (SSIM) (Wang et al., 2004), and Learned Perceptual Image Patch Similarity (LPIPS) (Zhang et al., 2018), to assess visual fidelity and perceptual quality. Then, we employ InstantMesh (Xu et al., 2024) to reconstruct 3D geometry from the generated views, and evaluate the resulting point clouds using Chamfer Distance (CD) and Intersection over Union (IoU) to quantify geometric consistency.

**Experimental Settings** We train ViewMask-1-to-3 using 24 H200 GPUs with a total batch size of 384 and a learning rate of $3\times10^{-5}$. A cosine learning rate scheduler is adopted, and training is performed under the Zero-2 optimization strategy. During inference, we set the sampling steps to 20.

### 4.1. Image to Multi-view Images

**Baseline** We compare our method against a comprehensive set of state-of-the-art diffusion-based novel view generation approaches, including Zero-1-to-3 (Liu et al., 2023b), Zero-1-to-3 XL (Liu et al., 2023b; Deitke et al., 2023), ViVid-1-to-3 (Kwak et al., 2024), Zero123++ (Shi et al., 2023), MV-Adapter (Huang et al., 2025), EpiDiff (Huang et al., 2024), TRELLIS (Xiang et al., 2025) and AR-1-to-3 (Zhang et al., 2025b).

**Qualitative Results** Fig. 4 presents a visual comparison between our method and several state-of-the-art multi-view generation approaches. In the top three rows, ViewMask-1-

*Table 1.* **Quantitative comparison on multi-view generation over the GSO and 3D-FUTURE datasets.** We report PSNR, SSIM, and LPIPS, ↑ indicates higher is better, ↓ indicates lower is better.

| Method | Architecture | GSO | | | 3D-FUTURE | | | Rank |
|---|---|---|---|---|---|---|---|---|
| | | PSNR↑ | SSIM↑ | LPIPS↓ | PSNR↑ | SSIM↑ | LPIPS↓ | |
| Zero-1-to-3 (Liu et al., 2023b) | 2D Cont. Diff. | 18.8219 | 0.8294 | 0.1659 | 17.0526 | 0.8163 | 0.1760 | 5.2 |
| Zero-1-to-3 XL (Liu et al., 2023b) | 2D Cont. Diff. | 19.6839 | 0.8381 | 0.1518 | 18.4702 | 0.8337 | 0.1485 | 3.0 |
| ViVid-1-to-3 (Kwak et al., 2024) | 2D Cont. Diff. | 19.7978 | **0.8566** | 0.1764 | 18.3241 | 0.8437 | 0.1682 | 3.3 |
| Zero123++ (Shi et al., 2023) | 2D Cont. Diff. | 19.6373 | 0.8045 | 0.3550 | **23.5001** | 0.8527 | 0.1529 | 3.8 |
| Epidiff (Huang et al., 2024) | 2D Cont. Diff. | 18.9917 | 0.8244 | 0.1667 | 17.4592 | 0.8108 | 0.1785 | 5.5 |
| MV-Adapter (Huang et al., 2025) | 2D Cont. Diff. | 19.4673 | 0.7518 | 0.1503 | 19.3375 | 0.7217 | 0.4036 | 6.5 |
| AR-1-to-3 (Zhang et al., 2025b) | Video Cont. Diff. | 13.5084 | 0.7376 | 0.3514 | 13.7724 | 0.7479 | 0.3034 | 7.3 |
| ViewMask-1-to-3 | 2D Disc. Diff. | **20.6112** | 0.8561 | **0.1497** | 19.9953 | **0.8650** | **0.1288** | **1.3** |

*Table 2.* **Quantitative comparison on 3D reconstruction over the GSO and 3D-FUTURE datasets.** We report chamfer distance (CD) value and IOU. ↑ indicates higher is better, ↓ indicates lower is better.

| Method | GSO | | 3D-FUTURE | |
|---|---|---|---|---|
| | CD↓ | IOU↑ | CD↓ | IOU↑ |
| Zero-1-to-3 | 0.0163 | 0.5665 | 0.0113 | 0.5005 |
| Zero-1-to-3 XL | 0.0159 | 0.5799 | 0.0106 | 0.5271 |
| Zero123++ | 0.0163 | 0.5832 | 0.0314 | 0.4250 |
| ViVid-1-to-3 | 0.0163 | 0.5841 | **0.0105** | 0.5246 |
| AR-1-to-3 | 0.0372 | 0.4380 | 0.0148 | 0.4286 |
| EpiDiff | 0.0166 | 0.5457 | 0.0121 | 0.4784 |
| TRELLIS | 0.0496 | 0.4026 | 0.0178 | 0.4396 |
| ViewMask-1-to-3 | **0.0149** | **0.5847** | 0.0106 | **0.5315** |

to-3 preserves accurate chair legs and seat curvature, avoiding the distortions and artifacts seen in other methods. In the bottom three rows, it achieves the best overall consistency, capturing both the recessed details on the desktop and the metal frame. Compared with these baselines, our method produces sharper visual quality, better preserves fine-grained textures, and maintains higher cross-view consistency.

**Quantitative Results** To quantitatively evaluate the zero-shot generalization capability of our model, we conduct experiments on two aspects: (1) *image-level quality*, measured on novel views generated from a single input image, and (2) *3D-level geometric accuracy*, evaluated on 3D reconstructions. These experiments are evaluated on the Google Scanned Objects (GSO) dataset (Downs et al., 2022) and the 3D-FUTURE dataset (Fu et al., 2021).

The input views are selected with controlled offsets in azimuth and elevation relative to the canonical frontal view of each 3D object. For models that support arbitrary-angle rendering, such as Zero-1-to-3 and Zero-1-to-3 XL, the views are rendered at the same angles as those used by ViewMask-1-to-3 to ensure a fair comparison. We then evaluate PSNR, SSIM, and LPIPS following each model's respective settings, and report the results in Tab. 1. Overall, our method, ViewMask-1-to-3, achieves the best combined

performance across both datasets and attains the highest rank (1.3), demonstrating its superiority in multi-view generation.

To evaluate geometric consistency, we reconstruct 3D shapes from the generated multi-view images for each model. We then compute Chamfer Distance (CD) and IoU between the reconstructed shapes and the ground-truth meshes. The quantitative results are reported in Tab. 2, demonstrating that our method achieves lower CD and higher IoU, indicating more accurate 3D reconstruction and better cross-view consistency.

### 4.2. Text to Multi-view Images

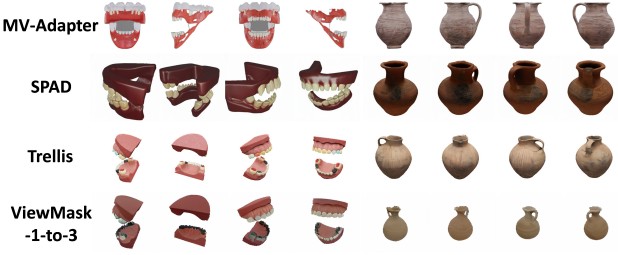

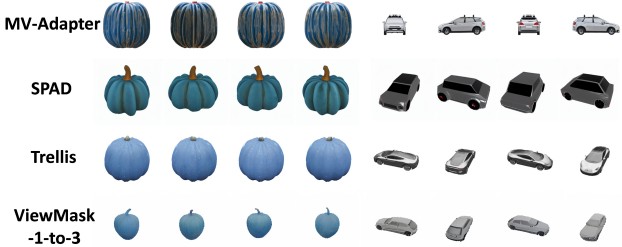

*Figure 5.* **Qualitative results on T2MV.** Examples of multi-view object generation from text prompts using our method, achieved directly without any intermediate steps.

In the T2MV task, we demonstrate the ability to gener-

ate multi-view images of an object directly from text inputs. Models like MV-Adapter (Huang et al., 2025) and TRELLIS (Xiang et al., 2025) support T2MV and I2MV using different checkpoints or architectures. Zero123++ (Shi et al., 2023) and ViVid-1-to-3 (Kwak et al., 2024) generate reference images with a T2I model before extracting the subject for novel views. In contrast, our method directly produces novel views that preserve both the specified viewpoint and the prompt's semantic content. As shown in Fig. 5, we present representative test samples from the Objaverse dataset and compare qualitative results with MV-Adapter (Huang et al., 2025), TRELLIS (Xiang et al., 2025), and SPAD (Kant et al., 2024), showing that our method achieves comparable visual quality and consistency.

### 4.3. Ablation study

**Mask Schedule** We investigate the impact of different mask scheduling strategies on our model's performance, including linear, quadratic, and cosine schedules.

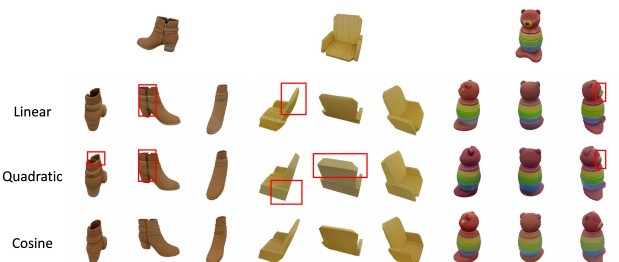

*Figure 6.* **Qualitative results of different mask scheduling strategies on a subset of the GSO dataset.** Compared with linear, quadratic, and fixed schedules, the cosine schedule produces sharper details and more coherent structures in the generated novel views.

In Tab. 3, our experiments on a subset of the GSO dataset indicate that the cosine schedule yields smoother optimization and superior final results compared with the other strategies, highlighting the importance of gradually adjusting the masking ratio over timesteps. In addition to quantitative evaluation, we provide qualitative visualizations of the generated images under different schedules, as shown in Fig. 6. For the boot, both the linear and quadratic schedules hallucinate an additional zipper on the opposite side, which is clearly inconsistent with reality. For the chair, the quadratic schedule yields incorrect thickness estimation. For the toy, both the linear and quadratic schedules distort the teddy bear's facial region.

### 4.4. Scalability Analysis

To further validate the scalability of our framework, we extend the original 4-view setting to higher-view generation scenarios, including 6-view and 8-view, using a subset of the original training dataset, as shown in Table 4. Build-

*Table 3.* **Ablation study on different mask scheduling strategies.** The metrics (e.g., PSNR, SSIM, LPIPS) are reported for each schedule.

| Schedule | PSNR ↑ | SSIM ↑ | LPIPS ↓ |
|---|---|---|---|
| Linear | 17.9882 | 0.8435 | 0.1882 |
| Quadratic | 17.9785 | 0.8431 | 0.1865 |
| Cosine | **18.0974** | **0.8437** | **0.1834** |

ing upon the 6-view setting, we further scale the model to 8-view generation. Notably, the 8-view setting exceeds the maximum token length used during training, thereby additionally demonstrating the model's ability to generalize beyond its original token budget. All evaluations are conducted on the GSO dataset.

*Table 4.* Scalability analysis under different numbers of generated views on the GSO dataset.

| Number of Views | PSNR ↑ | SSIM ↑ | LPIPS ↓ |
|---|---|---|---|
| 4 (baseline) | 20.6112 | 0.8561 | 0.1497 |
| 6 | 20.5956 | 0.8587 | 0.1405 |
| 8 | 20.5120 | 0.8561 | 0.1442 |

Transitioning from the original 4-view setting to 6-view and 8-view generation requires only 9k fine-tuning steps, indicating minimal adaptation overhead. Quantitative results show that increasing the number of generated views does not introduce noticeable performance degradation. On the contrary, the 6-view and 8-view settings maintain comparable, and in some cases slightly improved, reconstruction quality compared to the baseline. We attribute this partially to the fact that some additional target viewpoints are geometrically closer to the input view and are therefore easier to synthesize. Overall, these results demonstrate that our framework can scale effectively to higher-view generation while preserving generation quality and generalization capability.

### 4.5. Efficiency Analysis

To further evaluate the practical usability of the proposed framework, we analyze its inference efficiency and computational cost. As shown in Table 5, our model contains 8B parameters and performs inference with 20 sampling steps, achieving a total latency of 3.85s (1.28s/view). Despite the larger model scale, our method remains more efficient than continuous diffusion-based approaches, including ViVid-1-to-3 (5.99s/view) and AR-1-to-3 (5.35s/view). The model requires approximately 17GB GPU memory during inference, which is affordable on most modern high-performance GPUs. In exchange for this computational cost, the framework offers a unified and versatile architecture capable of handling multiple generation tasks within a single model,

*Table 5.* **Efficiency analysis of multi-view generation models.**

| Method | Architecture | Params (B) | New Views | Latency (s) | Latency/ View (s) | Memory (G) | Other Tasks | GSO PSNR↑ | SSIM↑ | LPIPS↓ |
|---|---|---|---|---|---|---|---|---|---|---|
| Zero-1-to-3 | 2D Cont. Diff. | 1.286 | 1 | 2.656 | 2.656 | 14.486 | - | 18.8219 | 0.8294 | 0.1659 |
| Zero123++ | 2D Cont. Diff. | 0.866 | 6 | 6.246 | **1.041** | **3.603** | - | 19.6373 | 0.8045 | 0.3550 |
| ViVid-1-to-3 | 2D Cont. Diff. | 4.835 | 4 | 23.977 | 5.994 | 10.216 | - | 19.7978 | **0.8566** | 0.1764 |
| MV-Adapter | 2D Cont. Diff. | 3.552 | 5 | 27.469 | 5.494 | 10.575 | T2MV (in different ckpt) | 19.4673 | 0.7518 | 0.1503 |
| AR-1-to-3 | Video Cont. Diff. | 0.886 | 6 | 32.116 | 5.353 | 4.627 | - | 13.5084 | 0.7376 | 0.3514 |
| ViewMask-1-to-3 | 2D Disc. Diff. | 8.082 | 3 | 3.849 | 1.283 | 17.022 | T2MV, Inpainting | **20.6112** | 0.8561 | **0.1497** |

including T2MV, I2MV, training-free inpainting, and 2D Turn-Style generation, while also naturally extending to T2I generation. These results demonstrate that discrete diffusion can achieve a favorable balance between generation quality, efficiency, and task generality under a unified paradigm.

### 4.6. Task Extension

Our framework can be readily extended beyond image-to-multi-view synthesis toward unified multimodal understanding and generation tasks, while also naturally supporting image inpainting and training-free 2D Turn-Style generation, as shown in Fig. 7.

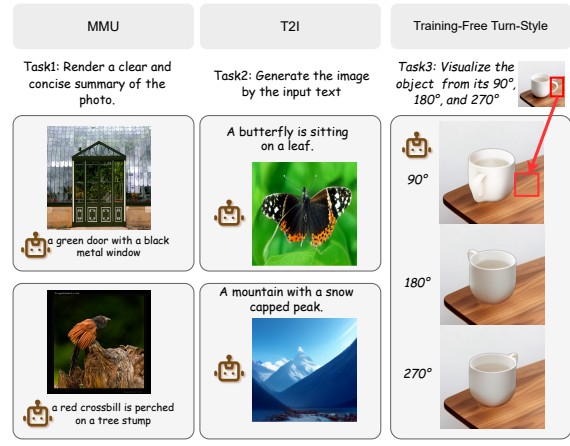

*Figure 7.* **Task Extension.** Left: basic multimodal understanding (MMU), e.g., captioning. Middle: text-to-image (T2I) generation without fine-tuning. Right: training-free 2D turn-style and completion preserving the background.

**Unified Multimodal Understanding and Generation** In addition to the aforementioned I2MV and T2MV capabilities, the Stage-1 pretraining endows our model with fundamental multimodal understanding and text-to-image generation abilities. Building on these abilities, our model naturally extends toward unified multimodal understanding and generation, evaluated on GenEval (Ghosh et al., 2023) for compositional text-to-image generation, as shown in Tab. 6. ViewMask-1-to-3 achieves the best overall score (0.72), outperforming unified models Show-o (0.68) (Xie et al., 2025b) and MMaDA (0.63) (Yang et al., 2026), and surpassing

generation-only models like SD3.5-L (0.71) (Esser et al., 2024), comparable to SANA-1.5 (0.72) (Xie et al., 2025a). (More discussions can be found in Appendix B.2)

**Training-Free 2D Turn-Style** Inspired by Adobe's Project Turn Style, which enables re-angling and rotating 2D illustrations as if they were 3D objects, we extend this capability in a *training-free* manner based on our ViewMask-1-to-3 framework. Benefiting from the inherent properties of discrete diffusion, our model can rotate a specified object by a given angle and perform image completion while preserving the original background, all without additional fine-tuning, as these abilities are implicitly embedded during diffusion training. These cases show that our framework extends to a unified diffusion architecture while maintaining strong flexibility and generalization across diverse tasks.

## 5. Conclusion

In this work, we introduce **ViewMask-1-to-3**, a discrete diffusion framework for multi-view image generation. Experiments on GSO and 3D-FUTURE demonstrate its strong performance, showcasing discrete diffusion as an effective and flexible approach for multi-view synthesis. Our results also highlight its potential for MMU, T2I, T2MV, and training-free 2D Turn-Style, paving the way toward more unified and scalable multi-view generation.

Despite strong performance, there is room to expand viewpoint diversity and improve lighting robustness. Future work will focus on enhancing scalability, resolution, and generalization for more unified multi-view generation.

## Acknowledgements

This paper is partially supported by the National Key R&D Program of China No.2022ZD0161000, and was completed during an internship at TeleAI.

## Impact Statement

This paper presents work whose goal is to advance the field of machine learning. There are many potential societal consequences of our work, none of which we feel must be specifically highlighted here.

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

# A. Additional Experimental Details

## A.1. Baseline Setting

Most of the baselines we select—such as Zero-1-to-3 (Liu et al., 2023b), Zero-1-to-3 XL (Liu et al., 2023b; Deitke et al., 2023), ViVid-1-to-3 (Kwak et al., 2024), Zero123++ (Shi et al., 2023), EpiDiff (Huang et al., 2024), AR-1-to-3 (Zhang et al., 2025b), TRELLIS (Xiang et al., 2025), and our own method ViewMask-1-to-3 are trained on data rendered with perspective projection. Among all these methods, only MV-Adapter (Huang et al., 2025) is trained using data rendered with orthographic projection.

## A.2. Ground-Truth Rendering

For Zero123++ and AR-1-to-3, the ground-truth images are rendered from 3D assets using their default six viewpoint configuration. MV-Adapter follows its original training protocol and renders four orthographic views consistent with its orthographic camera model. For the remaining baselines that support arbitrary multi-view rendering, we use the same viewpoint and camera parameters as ours to ensure fair comparison.

For models such as Zero123++ that by default generate images with a gray background, we render the corresponding ground-truth views using the same background color when reporting PSNR, SSIM, and LPIPS. Although some implementations offer background removal via rembg, this often removes thin or delicate parts of the object as well, which in turn compromises the reliability of the baseline's quantitative results.

*Table 6.* **Evaluation on Image Generation Benchmarks (GenEval).** Our model achieves strong performance in unified models and is also comparable to methods among generation-only models.

| Model | Architecture | Single Obj. | Two Obj. | Counting | Colors | Position | Color Attr. | Overall |
|---|---|---|---|---|---|---|---|---|
| *Generation-Only* | | | | | | | | |
| SDv1.5 (Rombach et al., 2022) | Diffusion | 0.97 | 0.38 | 0.35 | 0.76 | 0.04 | 0.06 | 0.43 |
| SDv2.1 (Rombach et al., 2022) | Diffusion | 0.98 | 0.51 | 0.44 | 0.85 | 0.07 | 0.17 | 0.50 |
| SD3.5-L (Esser et al., 2024) | Diffusion | 0.98 | **0.89** | 0.73 | 0.83 | 0.59 | **0.65** | 0.71 |
| DALL-E 2 (Ramesh et al., 2022) | - | 0.94 | 0.66 | 0.49 | 0.77 | 0.10 | 0.19 | 0.52 |
| DALL-E 3 (Betker et al., 2023) | - | 0.96 | 0.87 | 0.47 | 0.83 | 0.43 | 0.45 | 0.67 |
| LlamaGen (Sun et al., 2024) | AR | 0.71 | 0.34 | 0.21 | 0.58 | 0.07 | 0.04 | 0.32 |
| FLUX.1 [Dev] (Labs, 2024) | Diffusion | 0.98 | 0.81 | 0.74 | 0.79 | 0.22 | 0.45 | 0.66 |
| SDXL (Podell et al., 2024) | Diffusion | 0.98 | 0.74 | 0.39 | 0.85 | 0.15 | 0.23 | 0.55 |
| OmniGen (Xiao et al., 2025) | Diffusion | 0.98 | 0.84 | 0.66 | 0.74 | 0.40 | 0.43 | 0.68 |
| SANA-1.5 (Xie et al., 2025a) | Diffusion | **0.99** | 0.85 | **0.77** | **0.87** | 0.34 | 0.54 | **0.72** |
| *Unified Understanding & Generation* | | | | | | | | |
| SEED-X (Ge et al., 2024) | AR | 0.97 | 0.58 | 0.26 | 0.80 | 0.19 | 0.14 | 0.49 |
| LWM (Liu et al., 2024a) | AR | 0.93 | 0.41 | 0.46 | 0.79 | 0.09 | 0.15 | 0.47 |
| Janus (Wu et al., 2025a) | AR | 0.97 | 0.68 | 0.30 | 0.84 | 0.46 | 0.42 | 0.61 |
| Show-o (Xie et al., 2025b) | AR+Diff. | 0.98 | 0.80 | 0.66 | 0.84 | 0.31 | 0.50 | 0.68 |
| VAR-GPT (Zhuang et al., 2025) | AR | 0.96 | 0.53 | 0.48 | 0.83 | 0.13 | 0.21 | 0.53 |
| MMaDA (Yang et al., 2026) | Discrete Diff. | **0.99** | 0.76 | 0.61 | 0.84 | 0.20 | 0.37 | 0.63 |
| ViewMask-1-to-3 | Discrete Diff. | **0.99** | 0.88 | 0.50 | 0.86 | **0.60** | 0.53 | **0.72** |

# B. Additional Results

## B.1. Image to Multi-view Images

We provide additional I2MV comparison results, as shown in Fig. 8. Our model demonstrates more accurate control over fine-grained details.

## B.2. Task Extension

**Unified Multimodal Understanding and Generation** During inference, ViewMask-1-to-3 employs task-specific input templates that leverage discrete tokens for both vision and language modalities. For multimodal understanding (MMU)

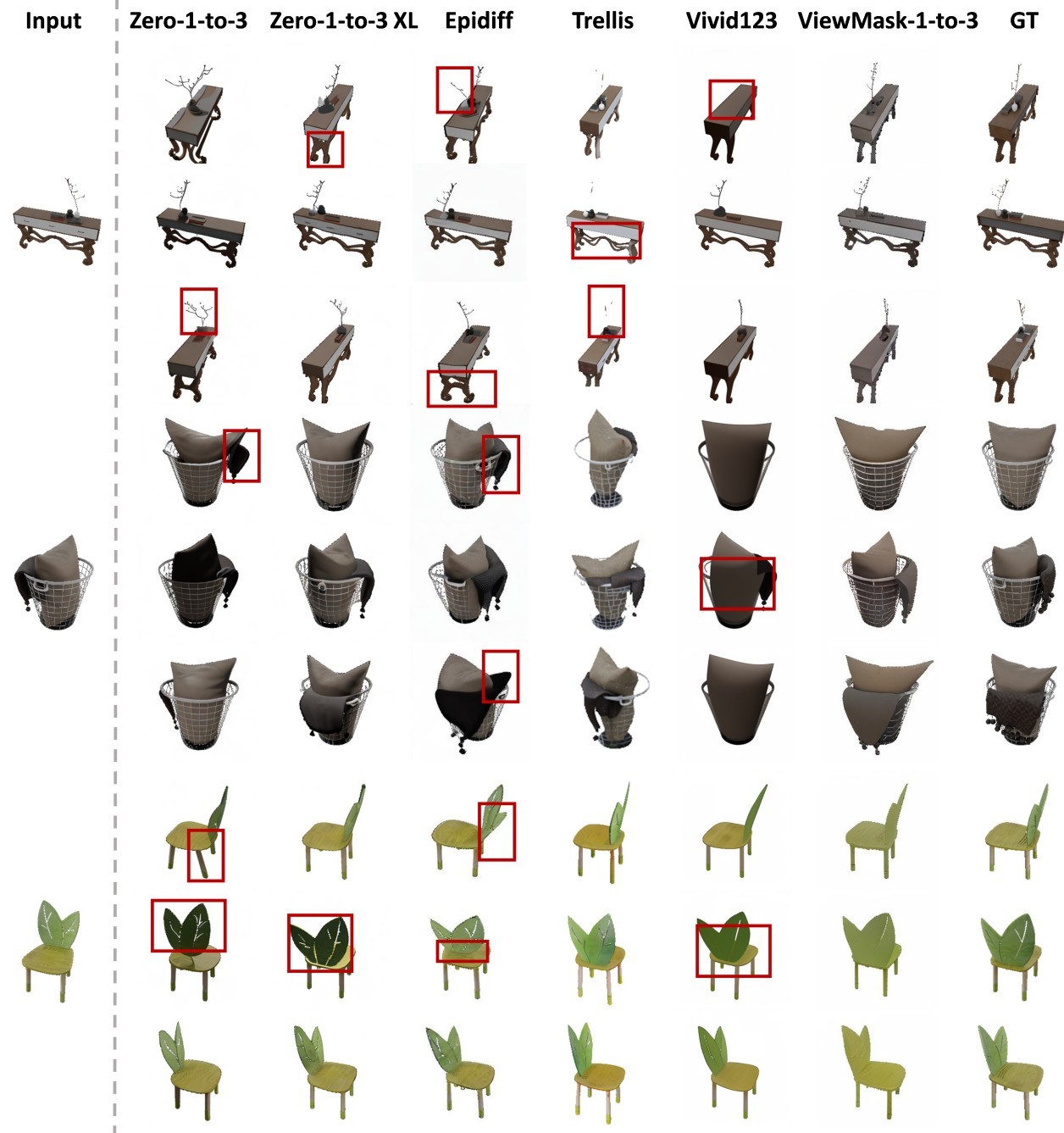

*Figure 8.* **Qualitative comparison on the 3D-Future dataset.** Visual results on the 3D-Future dataset, showing that our ViewMask-1-to-3 produces more consistent and realistic novel views compared with recent methods.

tasks, the input sequence is structured as:

[MMU] [SOT] *prompt* [EOT] [SOI] *image tokens* [EOI]

[SOT] *answer* [EOT],

where the text prompt and visual content are presented sequentially to guide the model's comprehension and response generation. The red-highlighted tokens correspond to the fully masked answer region, representing the content the model must produce.

For text-to-image (T2I) task, the sequence follows:

[T2I] [SOT] *caption* [EOT] [SOI] *image tokens* [EOI],

where the model autoregressively generates image tokens after the caption prompt.

We quantitatively evaluate the text-to-image generation capability of our model on the GenEval benchmark after 5k training steps on the BLIP-60K dataset, which evaluates object-centric generation under compositional prompts with diverse object attributes. Table 6 shows that ViewMask-1-to-3 achieves a strong overall score of 0.72, surpassing the specialized T2I model SD3.5-L and outperforming the architecturally similar unified model MMaDA by 0.09 (0.72 vs. 0.63), highlighting the practical potential of discrete diffusion architectures.

## C. Discrete Diffusion and Tokenization Choice.

**Advantages of Discrete Diffusion.** We adopt a discrete diffusion formulation for multi-view generation due to its favorable trade-offs in efficiency, architecture design, and functionality.

- **Efficiency.** Discrete diffusion models significantly improve inference efficiency compared to AR architectures. For instance, Lumina-DiMOO (Xin et al., 2025a) is reported to be $32\times$ **faster than** Lumina-mGPT 2.0 (Xin et al., 2025b).

- **Architecture.** Unlike autoregressive models with error accumulation, discrete diffusion uses **bidirectional attention over cross-view context** during training for iterative token refinement.

- **Functionality.** The masked prediction paradigm naturally supports flexible editing and completion tasks. In particular, it enables localized manipulation such as inpainting by conditioning on partial token observations.

*Table 7.* **Performance of discrete and continuous tokenizer on TokBench.** Show-O-MAGVIT2 achieves the best overall performance among discrete tokenizers across downsampling ratio, LFQ efficiency, video support, reconstruction quality, and codebook size.

| Type | Method | Train w/ Video Data | LFQ | Ratio | Codebook Size | rFID↓ | LPIPS↓ | PSNR↑ | SSIM↑ |
|------|--------|:---:|:---:|:---:|:---:|---|---|---|---|
| | Resize | - | - | $1\times$ | - | 5.39 | 0.06 | 27.71 | 0.84 |
| Discrete | TiTok | ✗ | ✗ | $1D$ | 4096 | 16.25 | 0.52 | 13.54 | 0.47 |
| | FlexTok | ✗ | ✗ | $1D$ | 64000 | 8.87 | 0.35 | 17.37 | 0.57 |
| | OmniTokenizer | ✓ | ✗ | $8\times$ | 8192 | 9.26 | 0.30 | 15.15 | 0.59 |
| | LlamaGen | ✗ | ✗ | $8\times$ | 16384 | 8.65 | 0.19 | 21.50 | 0.67 |
| | O-MAGVIT2 | ✓ | ✓ | $8\times$ | 262144 | 7.88 | **0.17** | **22.53** | **0.70** |
| | VQGAN | ✗ | ✗ | $16\times$ | 1024 | 12.63 | 0.36 | 17.29 | 0.55 |
| | Chameleon | ✗ | ✗ | $16\times$ | 8192 | 17.32 | 0.36 | 17.81 | 0.56 |
| | LlamaGen | ✗ | ✗ | $16\times$ | 16384 | 11.17 | 0.30 | 18.22 | 0.58 |
| | VAR | ✗ | ✗ | $16\times$ | 4096 | 8.91 | 0.24 | 19.98 | 0.63 |
| | MaskBit | ✗ | ✗ | $16\times$ | 16384 | 12.53 | 0.38 | 18.07 | 0.57 |
| | TokenFlow | ✗ | ✗ | $16\times$ | 32768 | 9.09 | 0.28 | 18.74 | 0.59 |
| | O-MAGVIT2 | ✓ | ✓ | $16\times$ | 262144 | 8.51 | 0.27 | 19.05 | 0.60 |
| | UniTok | ✗ | ✗ | $16\times$ | $8*4096$ | 7.82 | 0.20 | 21.15 | 0.66 |
| | MAGVIT-v2 | ✓ | ✓ | $16\times$ | 8192 | **7.40** | 0.18 | 18.47 | 0.60 |
| Continuous | DC-AE | ✗ | - | $32\times$ | - | 12.88 | 0.23 | 20.88 | 0.65 |
| | VA-VAE | ✗ | - | $16\times$ | - | 6.68 | 0.16 | 22.94 | 0.70 |
| | SD-XL | ✗ | - | $8\times$ | - | 7.60 | 0.19 | 22.52 | 0.69 |
| | SD-3.5 | ✗ | - | $8\times$ | - | 7.11 | 0.13 | 24.89 | 0.75 |
| | FLUX.1-dev | ✗ | - | $8\times$ | - | **6.42** | **0.11** | **25.50** | **0.77** |

**Tokenization Choice.** In Table 7, we evaluate several discrete tokenizers on TokBench (Wu et al., 2025b) and select MAGVIT-v2 as our default tokenizer, as it achieves the best overall trade-off between reconstruction fidelity and scalability.

Specifically, MAGVIT-v2 adopts a $16\times$ spatial downsampling factor for improved efficiency, employs LFQ quantization for better scalability, and is trained with image-video joint objectives, making it particularly suitable for multi-view consistency. Moreover, it maintains a well-balanced codebook and strong reconstruction quality, making it the most appropriate choice for our framework.

**Discussion on Discretization Error.** We acknowledge that discretization inevitably introduces reconstruction error and may limit the representation of high-frequency details due to finite codebook capacity. However, recent advances in neural tokenizers have significantly reduced this gap. Empirically, we observe that MAGVIT-v2 provides sufficiently accurate reconstruction for multi-view generation, and our results are comparable to or even outperform continuous baselines. This suggests that, in our setting, discretization is not a primary bottleneck, while its benefits in efficiency and structured generation outweigh the introduced quantization error.

