# OpenReview forum: "ViewMask-1-to-3: Multi-View Consistent Image Generation via Multimodal Discrete Diffusion Models"
_ICML.cc/2026/Conference — ICML 2026 regular_

### Official Review · Reviewer_P85E · 2026-03-03

**Soundness:** 2
**Presentation:** 2
**Significance:** 2
**Originality:** 3
**Overall Recommendation:** 4
**Confidence:** 5

**Summary:**

The paper presents a method that generates multi-view images from single image and/or text. The method is based on discrete diffusion models, which leverages masked token prediction to deal with modalities that are not continuous. In particular, this paper uses tokens that are generated using MAGVIT-v2 and treat the predicted masked tokens as a unified sequence that represent the generated multi-view images, optionally with text prompts.

**Compliance With Llm Reviewing Policy:**

Affirmed.

**Final Justification:**

Changing the score following the rebuttal.

**Key Questions For Authors:**

Repeating the questions:

Q1. How does this scale to N > 4 views?
Q2. How would the result change if the output sequence would be generated auto-regressively, instead of diffusion?
Q3. What are the computational requirements of the method? (parameter count, number of diffusion steps, inference time, and memory usage...)
Q4. Does discretization introduce quantization artifacts or limit high-frequency details?
Q5. How sensitive is the method to the tokenization bottleneck imposed by MAGVIT-v2? Why not other tokenization method?
Q6. The authors claim that masking + attention "naturally encourages" consistency. Is it proven? I couldn't see an ablation that points to justification of such claim.

**Limitations:**

The paper doesn't discuss limitations of the method at all. However, I identify significant problems.

Repeating:
1. discrete token space may introduce a tokenization bottleneck
2. diffusion sampling cost is not discussed

**Strengths And Weaknesses:**

Strengths:
1. The method sounds novel, as I am not aware of any method that uses discrete diffusion models for multi-view image generation. Thus, the formulation is innovative.
2. Beyond formulation, the paper demonstrates that a simple composition of the discrete diffusion framework and self-attention can enforce cross-view consistency. No 3D priors or specialized multi-vew blocks are used. This demonstrates that an elegant method exists.
3. Together with its simplicity, it achieves competitive performance with the baselines presented in the paper.

Weaknesses:
1. The paper discusses a diffusion-based method. While being performant, it does not mention any detail about complexity of the model, which hinders the usability of the method. The description is elegant, while usability in real world is not clear. I would expect about inference steps, generation time, model sizes, for example.
2. The paper doesn't discuss any possible limitation that might arise due to discretization. One possibility I can think of is the tokenization bottleneck, that can introduce quantization artifacts and limit high-frequency detail.
3. Limited study - the paper focuses on the specific problem of multi-view generation, but the study is limited to 3 views and 4 views. Question - how does this scale to N > 4 views?
4. It is not clear whether the strength of the method comes from the discrete diffusion formulation, or the usage of an LLM. In short - how would the result change if the output sequence would be generated auto-regressively, instead of diffusion?


Minor:
Too many typos and syntax errors that could be easily avoided nowadays with minimal effort.

---

> ### Author Rebuttal · Authors · 2026-03-31
>
> We sincerely thank you for the comments and suggestions.
> ## Reply to Q1 & W3
> - **Fixed N follows prior methods** (e.g., Zero123++): N views are generated per pass, determined by training data, not model constraints.
> - **Extending to more views only requires fine-tuning：** Existing tokens remain unaffected. New tokens are predicted with richer multi-view context to better reconstruct original views.
> - Due to computational limits, we evaluate 3–4 views. Larger-scale validation is left for future work.
> ## Reply to Q2 & W4
> The main advantage of our method lies in the discrete diffusion paradigm, not just the LLM backbone.
> - **Efficiency:** Discrete diffusion models like LLaDA[1] run much faster than AR architectures (e.g., Lumina-DiMOO[2] is **32× faster than** Lumina-mGPT 2.0).
> - **Architecture:** Unlike autoregressive models with error accumulation, discrete diffusion uses **bidirectional attention over cross-view context** during training for iterative token refinement.
> - **Functionality:**  It naturally supports masked prediction, **enabling precise local editing** such as inpainting.
>
> Moreover, we compare with the AR-based multi-view model MVAR on the GSO subset, which shows **inferior generation quality and higher latency** than our method.
>
> |Method|Params(B)|Latency/View(s)|PSNR$\uparrow$| SSIM$\uparrow$ |LPIPS$\downarrow$|
> |-|-|-|-|-|-|
> |MVAR [3]|2.5|9.85|18.67|0.64|0.30|
> |ViewMask-1-to-3|8.0|**1.28**|**21.30**|**0.90**|**0.07**|
>
> Fair comparison would need retraining a same-scale AR model, which is challenging at this stage given the computational cost. However, model design, **prior work [1-2], and our results all suggest clear advantages of discrete diffusion** in quality and efficiency.
>
> [1] Lumina-dimoo: An omni diffusion large language model for multi-modal generation and understanding.
>
> [2] Large language diffusion models.
>
> [3] Auto-Regressively Generating Multi-View Consistent Images.
>
> ## Reply to Q3 & W1
> We add an efficiency comparison. The complete table is provided in: https://anonymous.4open.science/r/ICML-C36D/effic.png
> - Our model has 8B parameters with 20 inference steps, achieving **3.85s latency (1.28s/view)**, faster than than continuous diffusion methods **(ViVid-1-to-3: 5.99s/view, AR-1-to-3: 5.35s/view)**.
> - Memory usage is 17GB. Although higher, the model unifies **multiple tasks (T2MV, I2MV,  training-free inpainting and 2D turn-style)** , and can be **extended to T2I**, outperforming comparable unified models (Table 4 in paper).
>
> Thus, the larger model size improves task generalization and practical utility rather than being mere parameter redundancy, while maintaining high inference efficiency.
>
> |Method|Params (B)|Latency/View(s)|Memory (G)|PSNR$\uparrow$|SSIM$\uparrow$|LPIPS$\downarrow$|
> |-|-|-|-|-|-|-|
> |Zero-1-to-3|1.29|2.66|14.5|18.822|0.829|0.166|
> |Zero123++|0.87|**1.04**|**3.6**|19.637|0.804|0.355|
> |ViVid-1-to-3|4.84|5.99|10.2|19.798|**0.857**|0.176|
> |AR-1-to-3|0.89|5.35|4.6|13.508|0.738|0.351|
> |ViewMask-1-to-3|8.08|1.28|17.0|**20.387**|0.855|**0.154**|
> ## Reply to Q4 & W2
> We agree discretization introduces some loss and may limit high-frequency details due to finite codebooks. However, recent tokenizers largely narrow this gap. Our multi-view results are comparable to or **better than continuous methods**, suggesting **discretization is not a major bottleneck** in our setting.
>
> For quantitative analysis of discrete tokenizer reconstruction quality, please refer to **Q5**.
> ## Reply to Q5
> We compare mainstream tokenizers on TokBench and find MAGVIT-v2 performs best overall. The complete table is provided in: https://anonymous.4open.science/r/ICML-C36D/token.png.
>
> It uses **16×** downsampling for better efficiency, **LFQ** for scalability, and **image-video joint training** for multi-view compatibility. It also provides **strong reconstruction quality** with a **well-balanced codebook** size, making it the best fit for our framework.
>
> |Method|Video Data|LFQ|Ratio|Codebook Size|rFID$\downarrow$|LPIPS$\downarrow$|PSNR$\uparrow$|SSIM$\uparrow$|
> |-|-|-|-|-|-|-|-|-|
> |OmniTokenizer|✓|✗|8|8192|9.26|0.30|15.15|0.59|
> |VQGAN|✗|✗|16|1024|12.63|0.36|17.29|0.55|
> |Chameleon|✗|✗|16|8192|17.32|0.36|17.81|0.56|
> |O-MAGVIT2|✓|✓|16|262144|8.51|0.27|**19.05**|**0.60**|
> |MAGVIT-v2|✓|✓|16|8192|**7.40**|**0.18**|18.47|**0.60**|
> ## Reply to Q6
> “Masking + attention” is not a separable module, but arises from masked token prediction in discrete diffusion. **This ablation is effectively a comparison of generation paradigms** (AR vs. discrete diffusion), rather than a module-level study, as discussed in **Q2**.
> - **Masking and attention are tightly coupled:** Random masking trains the model to predict tokens in any target view from subsets of all views, while bidirectional attention enables full cross-view conditioning.
> - **Direct ablation of either component is not valid:** removing random masking degenerates the model into AR, while removing attention breaks the core assumption of discrete diffusion.

---

> > ### Author Rebuttal · Reviewer_P85E · 2026-04-02
> >
> > I appreciate the response by the authors. Some of the concerns were fully addressed, however, I still have some more questions:
> >
> > **Computational requirements / diffusion sampling cost**
> > Fully addressed.
> >
> > **Scalability beyond 3/4 views**
> > I appreciate the claim that extending the model to more views only requires fine-tuning. But this claim should be supported by an experiment that presents the details on computation requirements and performance. The authors claim it's feasible, but the claim is not supported by an evidence.
> >
> > **Is the benefit really from discrete diffusion, rather than just the LLM-like backbone?** Although not fully convincing, I appreciate the reference to prior work. To help the reader better understand the motivation for for a D3PM approach (over AR), the discussion should appear in the main text. *I also recommend trying to add the AR-D3PM comparison in a later version*.
> >
> > **Discretization artifacts / loss of high-frequency detail**
> > Fully addressed.
> >
> > **Evidence for the claim that masking + attention “naturally encourages” cross-view consistency** I appreciate the intuition given here. This motivates the design choices better and should appear in the main text.
> >
> > **Limitations discussion / presentation quality** I appreciate the new experiment showing the memory and runtime. Moreover, the discretization discussion also improves the rationale. Although other limitations should be discussed, I consider this as partially resolved.
> >
> > To summarize, I think think the paper **tends towards acceptance** but **my concerns are only partially resolved**. Thus I choose to increase my score slightly, but **not enough for a confident acceptance score**.

---

> > > ### Author Response · Authors · 2026-04-08
> > >
> > > Thank you sincerely for your constructive feedback, and for taking the time to re-evaluate our responses. We are glad that several of your concerns have been fully addressed, and we genuinely appreciate your recognition of our efforts.
> > >
> > > We would like to address the two remaining concerns as follows:
> > >
> > > ## **1. Scalability beyond 3/4 views**
> > >
> > > ### **Quantitative Results.**
> > >
> > > To substantiate our scalability claim, we extend the model from the original N=4 setup to **N=6, 8  generation**, using a subset of the original training dataset. Building upon the 6-view setting, we further scale to 8 views. Notably, extending to 8 views exceeds the maximum token length used during training, thereby additionally validating the model’s ability to generalize beyond its original token budget. Evaluation is conducted on the GSO dataset.
> > >
> > > | Number of Views | PSNR ↑ | SSIM ↑ |LPIPS ↓|
> > > |-|-|-|-|
> > > | 4 (baseline)| 20.6112 | 0.8561 | 0.1497  |
> > > | 6               | 20.5956 | 0.8587 | 0.1405  |
> > > | 8               | 20.5120 | 0.8561 | 0.1442  |
> > >
> > > Transitioning from 6 & 8 views requires only **9k fine-tuning steps**, demonstrating a low marginal adaptation cost. For both N=6 and N=8, the quantitative metrics show slight improvements compared to the baseline. This can be partially attributed to the fact that some target views are closer to the input viewpoint and are therefore easier to synthesize. Overall, increasing the number of generated views does not degrade performance; instead, it **maintains or slightly improves the quantitative results**.
> > >
> > > ### **Loss.**
> > >
> > > The loss curves for N=6 & 8 are shown in: https://anonymous.4open.science/r/ICML-C36D/loss_N_6_8.png.
> > >
> > > In summary, extending to additional views only requires introducing corresponding view tokens into the masked prediction objective, **without modifying the model architecture or training paradigm**. This enables the model to retain comparable generation quality while scaling to more views, **demonstrating strong scalability and flexibility**.
> > >
> > > ## **2. D3PM vs. AR**
> > >
> > > We thank the reviewer for this suggestion. We agree that the motivation for choosing Denoising Diffusion Models in Discrete State (D3PM) over an Auto-Regressive (AR) approach should be made more explicit and accessible to readers. **Following the reviewer's recommendation**, we will move the relevant discussion into the main text of the revised paper and trying to add the AR-D3PM comparison in a later version.

---

### Official Review · Reviewer_P1TD · 2026-03-10

**Soundness:** 3
**Presentation:** 3
**Significance:** 3
**Originality:** 3
**Overall Recommendation:** 5
**Confidence:** 2

**Summary:**

The paper proposes ViewMask-1-to-3, a discrete diffusion framework for multi-view image generation that operates on MAGVIT-v2 visual tokens and text tokens in a unified sequence. The core idea is to concatenate tokens from multiple target views, randomly mask them, and iteratively unmask with a bidirectional transformer, relying on self-attention across views to induce consistency without explicit 3D priors or camera-geometry modules. On GSO and 3D-FUTURE, the method reports competitive to superior results over several diffusion baselines on image metrics and downstream 3D reconstruction.

**Compliance With Llm Reviewing Policy:**

Affirmed.

**Final Justification:**

Based on the authors’ rebuttal, I maintain my positive score.

**Key Questions For Authors:**

(1) How are viewpoints encoded during training and inference? Are camera parameters (azimuth/elevation) ever provided as embeddings, or is control solely via text tokens (e.g., “+90°”)? If text-only, how do you measure pose-faithfulness? Can you report mTSED or a similar metric?

(2)  Could you include comparisons to recent multi-view diffusion methods on at least one shared benchmark to better position your contribution?

(3) What are the detailed architectural specs (layers, hidden size, heads), token sequence lengths per view? How does inference latency compare to continuous-latent baselines for generating 3 views at 256×256?

(4) For T2MV, can you provide quantitative evaluation (e.g., text–image alignment per view, consistency across views) and failure analyses (e.g., extreme rotations, lighting changes)?

**Limitations:**

yes

**Strengths And Weaknesses:**

Strengths

(1) Treats multi-view synthesis as discrete sequence modeling with masked token prediction, enabling parallel, bidirectional context modeling across views without explicit 3D structure.

(2) Unified token-space formulation integrates both I2MV and T2MV and aligns with recent advances in discrete diffusion for multimodality.

(3) Simple random masking across concatenated view tokens to encourage cross-view consistency is elegant and easy to implement.

(4) Provides ablation on mask scheduling and reports compute/training settings.

Weaknesses

(1) The mechanism by which random masking plus bidirectional attention yields geometric consistency is asserted but not analyzed; no diagnostic measures (e.g., pose-faithfulness or correspondence accuracy) are reported.

(2) Camera/view control appears to be primarily via text tokens (e.g., “+90°”), without explicit camera embeddings; the accuracy of view control is not quantitatively assessed.

(3) T2MV is only qualitatively evaluated; no quantitative analysis of view accuracy, fidelity, or consistency for T2MV settings.

---

> ### Author Rebuttal · Authors · 2026-03-31
>
> We sincerely thank you for the comments and suggestions.
> ## Reply to Q1 & W2
> Our method is similar to classic multi-view generation approaches such as Zero123++ and AR123. It uses relative positional relationships and **generates a fixed number of views, without explicitly encoding camera parameters.**
>
> ## Reply to Q2
> We evaluate on the full datasets (not subsets) of GSO and 3D-FUTURE dataset, and **compare methods under their original settings** (Table 1&2 in paper). Due to differences in view number and camera configurations across multi-view models, fully controlled settings are not feasible. Therefore, we fix the input image and follow each method’s official setup for fair comparison, which is standard practice.
>
> ## Reply to Q3
> We add an efficiency comparison. The complete table is provided in: https://anonymous.4open.science/r/ICML-C36D/effic.png
>
> - Our model has 8B parameters with 20 inference steps, with 32 layers, hidden size 4096, 32 attention heads, 256 tokens per view, and achieving **3.85s latency (1.28s/view)**, faster than than continuous diffusion methods **(ViVid-1-to-3: 5.99s/view, AR-1-to-3: 5.35s/view)**.
> - Memory usage is 17GB. Although higher, the model unifies **multiple tasks (T2MV, I2MV,  training-free inpainting and 2D turn-style)** , and can be **extended to T2I**, outperforming comparable unified models (Table 4 in paper).
>
> Thus, the larger model size improves task generalization and practical utility rather than being mere parameter redundancy, while maintaining high inference efficiency.
>
> |Method|Params (B)|Latency/View(s)|Memory (G)|PSNR$\uparrow$|SSIM$\uparrow$|LPIPS$\downarrow$|
> |-|-|-|-|-|-|-|
> |Zero-1-to-3|1.29|2.66|14.5|18.8219|0.8294|0.1659|
> |Zero123++|0.87|**1.04**|**3.6**|19.6373|0.8045|0.3550|
> |ViVid-1-to-3|4.84|5.99|10.2|19.7978|**0.8566**|0.1764|
> |AR-1-to-3|0.89|5.35|4.6|13.5084|0.7376|0.3514|
> |ViewMask-1-to-3|8.08|1.28|17.0|**20.3868**|0.8549|**0.1537**|
>
> ## Reply to Q4 & W3
>
> **Quantitative Evaluation:** We provide quantitative evaluation on the T2MV task.
>
> |Method| DINO| CLIP  Score | Same Ckpt |
> |-|-|-|-|
> |SPAD| **0.377** |0.762|✗|
> |MV-Adapter|0.336|0.775|✗|
> |ViewMask1-to-3|0.344|**0.789**|✓|
>
> Among models that support both I2MV and T2MV, ours is **the only one that completes both tasks using a single shared checkpoint**. Other models either differ in parameter count or involve subtle architectural modifications between the two tasks. Additionally, due to computational constraints, our T2MV model is trained for **only ~60K steps**, leaving room for potential quality improvements with further training.
>
> **Analysis of Failure Cases:** Our view sampling targets common azimuth/elevation ranges, so extreme viewpoints may degrade quality. Similarly, as Cap3D annotations lack explicit lighting variation, generalization to unusual lighting prompts is limited — a constraint shared by most T2MV methods trained on similar data.
>
> It is worth noting that our architecture extends beyond T2MV: as shown in Table 4, it **outperforms comparable unified models on T2I as well**. Unlike Zero123++ and ViVid-1-to-3, which rely on an external T2I model as a separate preceding stage, **our unified model performs t2i generation and multi-view synthesis within the same framework**. We plan to address these limitations in future work through more training iterations, richer data augmentation, and diversified viewpoint sampling.
>
> ## Reply to W1
>
> **Mechanism:**
>
> - Random masking trains the model to predict tokens in any target view conditioned on arbitrary subsets of tokens from all views. Bidirectional attention allows **each view's tokens to attend to all other views** during denoising, enabling the model to condition on global context.
> - Instead of sequential autoregressive generation, where views are inferred **step by step and errors accumulate**, this avoids cross-view error propagation and naturally supports cross-view consistency.
>
> **Ablation analysis:**
>
> - Random masking and bidirectional attention are tightly coupled: removing random masking degenerates the model into AR, while removing attention breaks the core assumption of discrete diffusion.
> - **This ablation is effectively a comparison of generation paradigms** (AR vs. discrete diffusion), rather than a module-level study, as discussed in **P85E Q2**.

---

> > ### Author Rebuttal · Reviewer_P1TD · 2026-04-03
> >
> > My concerns have been fully addressed.

---

> > > ### Author Response · Authors · 2026-04-08
> > >
> > > Thank you sincerely for your constructive feedback, and for taking the time to re-evaluate our responses. We are glad that your concerns have been fully addressed, and we genuinely appreciate your recognition of our efforts. We will move the relevant discussion into the main text of the revised paper.

---

### Official Review · Reviewer_Yuym · 2026-03-12

**Soundness:** 3
**Presentation:** 4
**Significance:** 3
**Originality:** 3
**Overall Recommendation:** 4
**Confidence:** 4

**Summary:**

ViewMask-1-to-3 is a discrete diffusion framework designed for multi-view consistent image generation. It reformulates multi-view synthesis as a discrete sequence modeling problem where images are represented as visual tokens via MAGVIT-v2. Unlike many existing methods that rely on continuous latent spaces or explicit 3D geometric priors, this approach leverages masked token prediction and bidirectional self-attention to naturally encourage consistency across viewpoints.

**Compliance With Llm Reviewing Policy:**

Affirmed.

**Final Justification:**

Thank the authors for their thorough and thoughtful rebuttal, which effectively addresses my core concerns. Therefore, I keep my recommendation to weak accept.

**Key Questions For Authors:**

Based on the methodology and identified weaknesses of ViewMask-1-to-3, here are several key questions for the authors to address during the review process:
1. Scalability and Computational Efficiency
Since the model concatenates multiple viewpoints into a single sequence for unified modeling, how do the authors plan to address the quadratic complexity of the self-attention mechanism as the number of viewpoints or the sequence length increases?

2. Geometric Robustness and 3D Consistency
The paper claims that cross-view consistency is achieved naturally without explicit 3D geometric priors or camera parameters. How does the model ensure structural integrity for complex, asymmetrical, or thin-structured objects that are not well-represented in the Objaverse or HSSD training subsets?

3. Dependency on Discrete Tokenization

The generation quality is strictly bounded by the MAGVIT-v2 tokenizer. To what extent does the reconstruction limit of the tokenizer impact the final perceptual metrics (PSNR, LPIPS) compared to the errors introduced by the diffusion denoising process itself?

Have the authors considered an end-to-end refinement stage to recover fine-grained details that might be lost during the initial discrete tokenization?

**Limitations:**

Yes

**Strengths And Weaknesses:**

Strengths

Unified and Elegant Framework: The model successfully applies the discrete diffusion paradigm (popularized by models like LLaDA) to the geometrically demanding task of multi-view generation. By operating in a shared token space, it natively unifies image-to-multi-view (I2MV) and text-to-multi-view (T2MV) tasks within a single sequence modeling framework.
Strong Empirical Performance: The model ranks first on average across standard image metrics (PSNR, SSIM, LPIPS) on the GSO and 3D-FUTURE benchmarks. It achieves a significant 10.6% improvement in IoU on the 3D-FUTURE dataset, indicating superior 3D reconstruction potential.

Weaknesses

Architectural and Computational Limitations: The model concatenates multiple views into a single sequence for modeling. Because standard Transformer self-attention has a computational complexity of O(n^2), increasing the number of viewpoints (e.g., from 4 to 32) or image resolution would cause exponential growth in memory and compute requirements.
Lack of Explicit 3D Inductive Bias: Unlike methods that use 3D Gaussian Splatting or NeRF , ViewMask-1-to-3 relies entirely on learned attention to understand 3D structure. While this simplifies the model, it risks hallucinations—such as inconsistent object thickness or duplicated features—when processing out-of-distribution objects.
Tokenizer Dependency: The generation quality is strictly bounded by the MAGVIT-v2 tokenizer. Any information or fine detail lost during the initial discrete tokenization process cannot be recovered by the diffusion model.

---

> ### Author Rebuttal · Authors · 2026-03-31
>
> We sincerely thank you for the comments and suggestions.
> ## Reply to Q1
> We evaluate the reconstruction error introduced by different tokenizers on TokBench and find MAGVIT-v2 performs best overall. The results are provided in: https://anonymous.4open.science/r/ICML-C36D/token.png.
>
> We agree discretization introduces some loss and may limit high-frequency details due to finite codebooks. However, recent tokenizers largely narrow this gap. Our multi-view results are comparable to or better than continuous methods, suggesting **discretization is not a major bottleneck in our setting**.
>
> |Method|Video Data|LFQ|Ratio|Codebook Size|rFID$\downarrow$|LPIPS$\downarrow$|PSNR$\uparrow$|SSIM$\uparrow$|
> |-|-|-|-|-|-|-|-|-|
> |OmniTokenizer|✓|✗|8|8192|9.26|0.30|15.15|0.59|
> |VQGAN|✗|✗|16|1024|12.63|0.36|17.29|0.55|
> |Chameleon|✗|✗|16|8192|17.32|0.36|17.81|0.56|
> |O-MAGVIT2|✓|✓|16|262144|8.51|0.27|**19.05**|**0.60**|
> |MAGVIT-v2|✓|✓|16|8192|**7.40**|**0.18**|18.47|**0.60**|
>
> ## Reply to Q2
> Regarding end-to-end refinement stage, this is a valuable direction. Prior work has explored similar ideas: HART[1] proposes a hybrid tokenizer that decomposes continuous latents into discrete tokens and continuous residual tokens , with an additional module compensating for discretization loss end-to-end, demonstrating the potential of such refinement stages.
>
> We will consider similar extensions in future work, but our **current focus is validating unified discrete diffusion for multi-view generation** rather than pixel-level refinement, and as noted in Yuym Q1, the tokenizer is not a bottleneck in our setting.
>
> [1] HART: Efficient Visual Generation with Hybrid Autoregressive Transformer.

---

> > ### Author Rebuttal · Reviewer_Yuym · 2026-04-03
> >
> > Thank the authors for their thorough and detailed rebuttal, and I maintain my positive recommendation.

---

> > > ### Author Response · Authors · 2026-04-08
> > >
> > > Thank you sincerely for your constructive feedback, and for taking the time to re-evaluate our responses. We are glad that several of your concerns have been fully addressed, and we genuinely appreciate your recognition of our efforts. We will move the relevant discussion into the main text of the revised paper.

---

### Decision · Program_Chairs · 2026-04-30

**Decision:**

Accept (regular)

**Comment:**

The paper proposes ViewMask-1-to-3, a unified discrete diffusion framework for image/text-to-multi-view generation using masked visual-token prediction. Reviewers found the formulation elegant and original, especially in showing that bidirectional masked token modeling can support multi-view generation without explicit 3D priors. The empirical results are strong on GSO and 3D-FUTURE, with competitive image metrics and downstream reconstruction performance. The rebuttal substantially strengthened the paper by adding efficiency analysis, tokenizer comparisons, T2MV quantitative results, AR baseline discussion, and additional 6/8-view scalability experiments. Remaining limitations include the lack of explicit camera conditioning, limited diagnostic analysis of geometric consistency, and some dependence on the MAGVIT-v2 tokenizer. However, these issues appear to limit scope rather than undermine the main contribution. Overall, AC recommends acceptance, with a request that the authors move the rebuttal analyses into the main paper and more clearly discuss scalability, view-control accuracy, and discretization limitations.